# Cannabinoid Receptor 2 Alters Social Memory and Microglial Activity in an Age-Dependent Manner

**DOI:** 10.3390/molecules26195984

**Published:** 2021-10-02

**Authors:** Joanna Agnieszka Komorowska-Müller, Tanushka Rana, Bolanle Fatimat Olabiyi, Andreas Zimmer, Anne-Caroline Schmöle

**Affiliations:** 1Institute for Molecular Psychiatry, Medical Faculty, University of Bonn, 53127 Bonn, Germany; jkimp@uni-bonn.de (J.A.K.-M.); tanushka.rana@mdc-berlin.de (T.R.); olabiyi@uni-bonn.de (B.F.O.); anne.schmoele@uni-bonn.de (A.-C.S.); 2International Max Planck Research School for Brain and Behavior, Ludwig-Erhard-Allee 2, 53175 Bonn, Germany

**Keywords:** cannabinoid receptor 2 (CB_2_R), microglia, inflammaging, memory, lipofuscin

## Abstract

Physiological brain aging is characterized by gradual, substantial changes in cognitive ability, accompanied by chronic activation of the neural immune system. This form of inflammation, termed inflammaging, in the central nervous system is primarily enacted through microglia, the resident immune cells. The endocannabinoid system, and particularly the cannabinoid receptor 2 (CB_2_R), is a major regulator of the activity of microglia and is upregulated under inflammatory conditions. Here, we elucidated the role of the CB_2_R in physiological brain aging. We used CB_2_R^−/−^ mice of progressive ages in a behavioral test battery to assess social and spatial learning and memory. This was followed by detailed immunohistochemical analysis of microglial activity and morphology, and of the expression of pro-inflammatory cytokines in the hippocampus. CB_2_R deletion decreased social memory in young mice, but did not affect spatial memory. In fact, old CB_2_R^−/−^ mice had a slightly improved social memory, whereas in WT mice we detected an age-related cognitive decline. On a cellular level, CB_2_R deletion increased lipofuscin accumulation in microglia, but not in neurons. CB_2_R^−/−^ microglia showed an increase of activity markers Iba1 and CD68, and minor upregulation in *tnfa* and *il6* expression and downregulation of *ccl2* with age. This was accompanied by a change in morphology as CB_2_R^−/−^ microglia had smaller somas and lower polarity, with increased branching, cell volume, and tree length. We present that CB_2_Rs are involved in cognition and age-induced microglial activity, but may also be important for microglial activation itself.

## 1. Introduction

Inflammaging, low-grade age-dependent inflammation, has been named one of the seven pillars of aging [1,2,3] and is one of the main causes of altered intracellular communication. In this type of inflammation, accumulating molecular signals produced throughout life act as the primary stimuli that activate macrophages and microglia [2,3]. These molecular signals can include a dysfunctional immune system that fails to efficiently clean pathogens, enhanced pro-inflammatory tissue damage, cellular senescence, enhanced NF-kB activation, or a defective autophagy response [4].

In the brain, inflammaging affects the activity of the resident innate immune cells—microglia. In young mice, microglia scan their surroundings to react to changes in the environment. Upon detection of neuronal damage or assault, they travel to the site of injury to phagocytose debris and to potentially induce a neuroinflammatory signaling cascade. However, this process is disturbed with aging. Aged microglia are less motile and have deficits in their phagocytic capacity, but show increased secretion of pro-inflammatory cytokines [5]. This age-induced priming of microglia is thought to influence their responses to infections or even stress [6,7]. Aged microglia also frequently become senescent, which further hinders their protective functions [7].

Many studies indicated that the endocannabinoid system (ECS) is an important regulator of microglial activity [8,9,10] and age-related cellular and molecular changes. ECS consists of two main receptors, the endocannabinoid receptors 1 and 2 (CB_1_R and CB_2_R); their ligands the endocannabinoids (ECs) 2-arachidonoylglycerol and anandamide, as well as EC-synthesizing and -degrading enzymes. Presynaptic CB_1_Rs are an integral part of a synaptic feedback mechanism [11], whereas CB_2_Rs modulate immune cell functions and microglia activity. Under basal conditions, CB_2_R expression in the brain is low and not readily detectable with most conventional methods [12,13,14,15,16]. However, it is upregulated under inflammation [17]. Moreover, recent findings also support the presence of functional CB_2_Rs on neurons [13,18,19,20,21,22].

Mice lacking CB_1_R exhibit accelerated age-related cognitive decline, gliosis, and increased expression of inflammatory cytokines in the brain [23,24,25,26]. At the same time, overall endocannabinoid tone decreases with age, as 2-AG level and DAGLα expression declined in 12-month old versus 2-month old mice alongside with CB_1_R binding to G-protein [27,28,29]. A chronic low-dose treatment of 18-month old mice with Δ^9^-THC, a CB_1_/CB_2_ agonist, resulted in recovery of their cognitive impairment to the levels of 2-month old mice [30]. The change in cognition was accompanied by an increase in synaptic proteins and in dendritic spine density in the hippocampus [30]. While these data suggest that the ECS is an important player in brain aging, its precise function remains unclear. In particular, it remains unknown if and how CB_2_Rs contribute to brain aging.

In this study, we characterized the role of CB_2_R in physiological brain aging, focusing on cognition and inflammaging. We investigated cognitive performance of young, adult, and old CB_2_R^−/−^ mice, and subsequently analyzed age-induced changes in microglial morphology and activity.

## 2. Results

### 2.1. CB_2_R Deletion Has a Moderate Age-Dependent Effect on Cognition

To investigate the age-related cognitive performance in CB_2_R^−/−^ mice, we used the partner recognition (PR) and Morris water maze (MWM). Anxiety-related behaviors were analyzed in the o-maze test (Figure 1A). All experiments were performed with young (3-months), adult (12-months), and old (18-months) male mice.

In the PR test, all groups showed intact sociability (Figure 1B), as evidenced by a significantly increased preference for the caged mouse of the metal can (one sample-test **** *p* < 0.0001 for each group). We detected no significant effects of genotype or age with regard to sociability. WT mice recognized their previous partner after 30 min separation and showed a preference for the novel partner in the 3-months and 12-months, but not in the 18-months group (one sample *t*-test against a hypothetical mean (50%): 3-months WT mice *p* = 0.0468; 12-months WT mice *p* = 0.0035; 18-months old CB_2_R^−/−^ mice *p* = 0.0472) (Figure 1C). We also detected a significant decrease in preference between 3-months and 18-months indicating an age-related cognitive decline (two-way ANOVA age x genotype effect: F_2,71_ = 17; *p* = 0.0336). In contrast, the preference in CB_2_R^−/−^ mice was higher than the chance level exclusively in the 18-month group. Consistently, after 1 h separation, we determined a preference for the novel partner in 3-month, but not in 12-months or 18-months old WT mice (one sample *t*-test against a hypothetical mean (50%): 3-months WT mice *p* = 0.0021) (Appendix A). We also revealed a significant age-related decrease in preference between 3-months and 12-months and 3-months and 18-months WT mice. In comparison, preference of the CB_2_R^−/−^ did not differ from the chance level in any of the investigated age groups, but it was increased in the 18-months old group in comparison to WT mice. Thus, CB_2_R deletion caused a moderate age-dependent change in social memory.

In the MWM test, all age groups of CB_2_R^−/−^ mice and WT controls showed a similar improvement during the acquisition phase and a similar performance during the reversal phase of the test (RM ANOVA: 3-months acquisition time: F_5,135_ = 12.29; *p* < 0.0001, reversal time: F_2,54_ = 8.366; *p* = 0.0007; 12-months acquisition time: F_5,135_ = 33.06; *p* < 0.0001, reversal interaction: F_2,54_ = 3.92; *p* = 0.0257, time: F_2,54_ = 20.04; *p* < 0.0001, 18-months acquisition time: F_5,135_ = 31.62; *p* < 0.0001, reversal time: F_2,54_ = 12.82; *p* < 0.0001) (Figure 1D). Additionally, all groups showed preference for the target quadrant during the probe trial (one sample *t*-test against a hypothetical mean (22.5 s): WT mice: 3-months *p* = 0.0088; 12-months *p* = 0.0002; 18-months *p* = 0.0421; CB_2_R^−/−^ mice: 3-months *p* = 0.0219; 12-months *p* = 0.0008; 18-months *p* = 0.0018) (Appendix A). Moreover, changes in memory performance cannot be explained by changes in motility, as we did not detect any significant genotype effect in distance travelled or velocity in any of the tests (Appendix A). Taken together, these provide no evidence for age-related, CB_2_R-mediated effects on cognitive performance.

### 2.2. CB_2_R Deletion Decreases Anxiety in an Age-Independent Manner

In the o-maze test, we determined a significant age and genotype effect for the time spent in the open compartment (two-way ANOVA age effect: F_2,82_ = 10.26; *p* = 0.0001; genotype effect: F_1,82_ = 4.427; *p* = 0.0384) (Figure 2A). Post hoc testing showed that the time spent in the open compartments decreased significantly in adult and old mice. We did not detect any significant genotype differences within the same age-group (Appendix A). Furthermore, we measured a significant age and genotype effect for the distance travelled in the open compartment. We revealed a significant decrease between 3- and 18-month old WT mice and 3- and 12-months old as well as 3- and 18-months old CB_2_R^−/−^ mice (two-way ANOVA age effect: F_2,82_ = 17; *p* < 0.0001; genotype effect: F_1,82_ = 7.009; *p* = 0.0097). Post hoc analysis did not reveal any significant differences between genotypes within the same age group, but we noted a trend for increased distance travelled in the open compartment in CB_2_R^−/−^ mice.

Then, we assessed behaviors associated with anxiety and risk assessment (Figure 2B). An increased number of stretched postures and a decreased amount of head-dipping is interpreted as increased anxiety-like behavior. In contrast, we detected a genotype and age effects for the number of stretched postures (two-way ANOVA age effect: F_2,81_ = 4.152; *p* = 0.0192; genotype effect: F_1,81_ = 13.24; *p* = 0.0005). Additionally, we measured a genotype effect and a significant increase of head-dipping behavior in CB_2_R^−/−^ mice in all age groups (two-way ANOVA genotype effect: F_1,81_ = 33.24; *p* < 0.0001).

### 2.3. Age-Dependent Increase of Lipofuscin Affected by CB_2_R Deletion in Microglia, but Not in Neurons in the Hippocampus

We next measured the accumulation of lipofuscin in hippocampal pyramidal neurons as age-related lipofuscin accumulation is associated with neuronal loss [31].

The age-related accumulation of lipofuscin, as measured by the area covered and particle density (Figure 3A,B), was similar in WT and CB_2_R mice (two-way ANOVA area covered, age effect: F_2,66_ = 81.41, *p* < 0.0001, particle density age effect: F_2,65_ = 82.98, *p* < 0.0001) (Appendix A).

We next analyzed the accumulation of lipofuscin in microglia from hippocampal stratum radiatum as age-related lipofuscin accumulation is also associated with microglial functional decline [32,33].

Lipofuscin accumulation in hippocampal radial microglia as measured by the area covered increased significantly with age in both WT and CB_2_R^−/−^ (two-way ANOVA area covered, age effect: F_2,419_ = 47.06, *p* < 0.0001) (Figure 4A,B). In WT mice lipofuscin increased from on average 0.68% (3-months) to 1.06% (12-months) and reached 3.72% (18-months), while in microglia from CB_2_R^−/−^ mice increased from 0.21% (3-months) to 3.35% (12-months) and reached 4.4% (18-months). The age-related increase of lipofuscin-accumulation was higher in CB_2_R^−/−^ mice (two-way ANOVA area covered, genotype effect: F_1,419_ = 7.857, *p* = 0.0053) which resulted in an interaction effect (two-way ANOVA area covered F_2,419_ = 7.514; *p* = 0.0006) (Figure 4B). Aligned with the enhanced covered area, the particle density significantly increased with age in both genotypes (two-way ANOVA particle density, age effect: F_2,429_ = 48.34, *p* < 0.0001) (Figure 4C) but was not significantly different between WT and CB_2_R^−/−^ mice (Appendix A).

### 2.4. Age-Induced Microglial Activity Is Altered in CB_2_R^−/−^ Microglia

Next, we analyzed Iba1 intensity and CD68 area fraction in the somas of hippocampal radial microglia from WT and CB_2_R^−/−^ mice to characterize microglial activity.

Iba1 intensity increased in both WT and CB_2_R^−/−^ microglia with age (two-way ANOVA MGV, age effect: F_2,345_ = 11.43, *p* < 0.0001) (Figure 5A,B). This was accompanied by an age-induced increase in CD68 expression in both WT and CB_2_R^−/−^ microglia (two-way ANOVA area covered, age effect: F_2,334_ = 18.11, *p* < 0.0001) (Figure 5C). Interestingly, CB_2_R^−/−^ microglia showed significantly enhanced Iba1 intensity when compared with WT microglia (two-way ANOVA MGV, genotype effect: F_1,345_ = 36.48, *p* < 0.0001, interaction effect: F_2,345_ = 3.525, *p* = 0.0305) (Figure 5B). This was further supported by enhanced CD68 content in CB_2_R^−/−^ microglia from 18-months old mice (Figure 5C).

We also analyzed the expression of the inflammatory mediators: *tnfa*, *il6, ccl2, arg1* and *nos2* in the hippocampus as markers of inflammaging.

Expression of *tnfa* (two-way ANOVA age effect: F_2,28_ = 7.804, *p* = 0.002) (Figure 6A), *il6* (two-way ANOVA age effect: F_2,28_ = 15.08, *p* < 0.0001) (Figure 6B) and *ccl2* (two-way ANOVA interaction genotype × age effect: F_2,29_ = 4.738; *p =* 0.0166; age effect: F_2,29_ = 12.29; *p =* 0.0001) increased with age in the hippocampus. In WT mice, we observed a steady increase of *tnfa* expression. The expression of *il6* increased from 3- to 12- months but then decreased from 12- to 18- months, back to the *il6* expression levels at 3- months (Figure 6B).

The age-dependent increase of *tnfa* expression was more prominent in CB_2_R^−/−^ with significant increase between 3-months and 12-months and 3-months and 18-months (Figure 6A). Similarly, *il6* expression increased from 3- to 12- months, but in contrast to WT mice, it did not significantly decrease between 12-months and 18-months (Figure 6B). However, there was no significant difference between 3-months and 18-months old CB_2_R^−/−^ mice. Expression of *ccl2* increased significantly between 3- and 18-months and 12- and 18-months exclusively in WT mice (Figure 6C). This resulted in a lower expression of *ccl2* in 18-month old CB_2_R^−/−^ mice in comparison to WT controls.

The expression of *arg1* was not altered by aging or CB_2_R deletion (Figure 6D), whereas for the expression of *nos2,* we detected an age effect, but no genotype effect (Figure 6E; two-way ANOVA age effect: F_2,26_ = 4.275; *p =* 0.0248).

Thus, CB_2_R deletion did not majorly alter cytokine expression in the hippocampus, but subsided an age-related increase in the *ccl2* expression.

To characterize the role of CB_2_R microglial activation in the context of inflammaging, we analyzed 3D microglial morphology of WT and CB_2_R^−/−^ hippocampal microglia.

The soma size of microglia significantly increased with age in both WT and CB_2_R^−/−^ (soma size, age effect: F_2,1000_ = 22.97; *p* < 0.0001) (Figure 7B). However, the increase was significantly less prominent in CB_2_R^−/−^ microglia as we detected a significant decrease in soma size in microglia from 18-months CB_2_R^−/−^ old mice.

Volume, ramification index, polarity index, tree length, and average branch length (Figure 7C–G) were not significantly altered with age (Appendix A). In contrast, microglia from CB_2_R^−/−^ mice showed an altered morphology with significant differences in volume, ramification index, polarity index, tree length, and average branch length (genotype effect, soma size: F_1,1000_ = 15.99; *p* < 0.0001; volume: F_1,316_ = 11.6; *p* = 0.0007; ramification index: F_1,317_ = 15.18; *p* = 0.0001; polarity: F_1,340_ = 15.27; *p* = 0.0001; tree length: F_1,329_ = 13.02; *p* = 0.0004; average branch length: F_1,326_ = 5.847; *p* = 0.0161) (Figure 7C–G), which was most prominent at the age of 3 months.

## 3. Discussion

We report that CB_2_R deletion, in contrast to CB_1_R deletion, has little to no effect on age-related changes in cognitive or anxiety-related behaviors. Nevertheless, we detected subtle genotype effects on inflammaging and microglial function. CB_2_R^−/−^ microglia exhibited an increased age-related lipofuscin accumulation and enhanced Iba1 and CD68 levels. Molecular changes were accompanied by altered microglial morphology, and moderately changed secretion of proinflammatory cytokines.

Several studies have shown that administration of THC to old animals was able to reverse many of the adverse consequences of aging on brain physiology and cognitive functions [30,34,35]. Whereas it has been established that CB1 receptors are the main target for these pro-cognitive effects of THC, the involvement of CB2 is less clear. Our results now suggest that CB2 receptors have little influence on the age-related decline of cognitive functions. However, they modulate age-related changes in the brain’s inflammatory milieu and thus may be involved in the effects of THC on brain inflammaging.

Recent reports confirming the presence of functional CB_2_R on neurons prompted us to investigate if CB_2_R deletion also results in an accelerated aging phenotype with early cognitive impairment, similar to what has been observed in CB_1_R^−/−^ mice. We found no evidence for an accelerated age-dependent memory decline in CB_2_R^−/−^ mice. On the contrary, we observed that the changes of social memory were inversely correlated with age in CB_2_R^−/−^ mice. While young CB_2_R^−/−^ mice showed a slight decrease in social memory, old mice performed slightly better than age-matched controls. In agreement with the former, we have recently reported an impairment in social memory in CB_2_R^−/−^ mice of both sexes aged between 4 and 6-months [34]. In contrast, we did not observe any significant differences in long-term spatial memory, although we detected trends indicating a slightly better performance of 18-month CB_2_R^−/−^ mice. The impairment in social memory in young mice did not arise due to altered anxiety-like behaviors. Contrary to previous reports, we measured an age-independent decrease in anxiety-like behavior in CB_2_R^−/−^ mice [35].

In agreement with our findings, other studies done on young CB_2_R^−/−^ mice showed a decrease in hippocampus-dependent fear memory [36,37]. Synaptic changes might underlie the cognitive deficits that we and others observed in younger CB_2_R^−/−^ mice, as CB_2_R deletion decreased dendritic spine density in the hippocampus [36,38].

Previous studies have investigated CB_2_R age-related changes in the context of Alzheimer’s disease (AD), where modulating CB_2_R function impacted microglia activity, amyloid plaque load and cognitive abilities of AD-related model mice [39,40,41]. Whether cognitive changes were due to a direct modulation of the CB_2_R on neurons or through microglia activity regulation remains unclear. Likewise, it is possible that either or both neuronal and microglial CB_2_Rs contributed to the social memory phenotype that we observed in our study. Nevertheless, our study strengthens an important role of CB_2_R in microglia activity in the context of inflammaging as we observed that the deletion of the CB_2_R increased accumulation of lipofuscin and CD68 levels in aged microglia.

CB_2_R expression is increased in microglia and macrophages in many diseases and acute inflammatory states, but also during aging, which may be due to inflammaging [41,42]. Aged microglia frequently have dystrophic morphology, with increased pro-inflammatory and decreased neuroprotective functions [33,43,44,45]. Therefore, we investigated age-related microglial activation by assessing Iba1 expression and cell morphology. Microglia from young CB_2_R^−/−^ mice had slightly increased cell surface, processes tree-length, number of branches, and ramification index, while their polarity index was decreased. These morphological changes were in line with a less-reactive microglial state and became less pronounced with aging. Microglial soma size increased with age, which is in line with previous studies [46]. However, the somas of CB_2_R^−/−^ microglia were smaller than those of WT microglia—especially in old mice—which could indicate reduced microglial activity. The aforementioned morphological changes were consistent with a less reactive state of microglia observed in young CB_2_R^−/−^ mice. It is in agreement with previous findings showing a dampened immune response to a pro-inflammatory stimulus [40], thus indicating that CB_2_R signaling is required for an efficient microglia activation in young mice. Furthermore, a subtle decrease in an average branch length in addition to an increase in whole-cell Iba1 levels and CD68 levels, was detected in old mice. Age-related increases in Iba1 levels [47,48] and enhanced CD68 levels [49] were reported previously, supporting the idea that microglial reactivity in older CB_2_R^−/−^ mice was accompanied by increased phagocytic activity. However, we did not observe corresponding changes in microglial morphology that would indicate a more reactive microglial state. Thus, it is possible that in CB_2_R^−/−^ mice, the upregulation of Iba1 and CD68 did not result in a functional change of microglia.

One of the key characteristics of aging is a disturbed proteostasis, which can be observed by an intracellular accumulation of potentially damaging protein aggregates [4]. Among others, lipofuscin has been shown to accumulate both in neurons and microglia in an age-dependent manner and impair cognition [31,50,51,52]. One main hypothesis states that lipofuscin accumulation in microglia occurs due to an increased phagocytosis of neuronal debris [53,54]. This is supported by findings that accumulation of lipofuscin-like lysosomal particles in microglia is connected with increased phagocytosis of myelin and suggests microglial degradative pathways as a critical target [32]. This accumulation in turn possibly leads to impaired microglial functions [55]. A loss of CB_1_R accelerated lipofuscin accumulation in the hippocampus [56]. Therefore, we hypothesized that the age-dependent effect of CB_2_R deletion on cognition could be a result of altered lipofuscin accumulation in either neurons or microglia. We detected an age-related increase of lipofuscin in both WT and CB_2_R^−/−^ hippocampal microglia, as reported previously [32,33,56]. However, exclusively CB_2_R^−/−^ microglia showed enhanced lipofuscin accumulation, suggesting a deficit in lysosomal degradation. This idea is supported by a recent study that showed that CB_2_R activation promotes the autophagy flux in macrophages and that the autophagy-lysosome pathway was involved in CB_2_R-mediated HMGB1 (High mobility group box 1) degradation [57]. Nevertheless, the role of CB_2_R in lysosomal pathways is still not well understood. Future studies should therefore include a more detailed analysis as to whether CB_2_R deletion impacts microglial phagocytosis and autophagy and whether these processes are also modulated by enhanced lipofuscin-accumulation.

We also analyzed the expression of inflammatory mediators *tnfa*, *il6, ccl2, nos2*, and *arg1*. *Tnfa*, *ccl2*, and *il6* expression increased with age as reported previously [33,58], whereas *arg1* expression was not changed with age. In our study, *il6* expression increased up to the age of 12 months and then decreased again at the age of 18 months. Since astrocytes also produce *il6,* we cannot exclude the possibility that astrocytes might dilute the direct effects of microglia. Interestingly, the age-induced increase of *tnfa* expression was even more pronounced in CB_2_R^−/−^ mice when compared to WT mice. In contrast, *ccl2* expression was significantly lower in CB_2_R^−/−^ mice than in WT at the age of 18 months. We observed similar effects before in vitro and in an AD mouse [40], supporting our idea that CB_2_R deletion alters inflammaging.

CB_2_R mediated signaling in microglial response during aging was previously investigated only in the context of age-related neuroinflammatory diseases (including Alzheimer’s disease). CB_2_R activation decreased microglial activity [41,59,60]. Similar results were also observed in AD-related mouse models after CB_2_R deletion [40,41]. However, one should consider that pharmacological CB_2_R activation/inhibition represents acute effects, whereas CB_2_R deletion represents chronic effects, which might be highly variable, especially during long-term processes such as inflammaging. We have recently shown that CB_2_R is necessary for toll like receptor (TLR)-mediated microglial activation [61]. Stimulated microglia from CB_2_R^−/−^ mice had distinct gene expression patterns, disturbed downstream signaling, and failed to show morphological signs of reactivity [61]. The findings suggest that CB_2_R activation is not only able to shift microglial activity from a pro- to an anti-inflammatory state but is also necessary to induce microglial activation in general. These recent data suggest that the role of CB_2_R on microglial activation is crucial and significantly more complex than previously thought and therefore needs to be investigated more thoroughly.

Taken together, we report that CB_2_R deletion has no effects on long-term spatial memory but has mild effects on short-term social memory during aging. Furthermore, we showed an age-dependent increase of lipofuscin in CB_2_R^−/−^ microglia but not in CB_2_R^−/−^ neurons. Enhanced lipofuscin accumulation in CB_2_R^−/−^ hippocampal microglia was accompanied by increased Iba1 and CD68 levels. Microglial morphology was not majorly altered with age as aging exclusively increased microglia soma size but did not alter other investigated parameters. However, CB_2_R^−/−^ microglia showed morphological differences independent of age with increased cell volume, ramification index, and process tree length, and decreased polarity and soma size. We conclude that CB_2_R plays a role in cognition and microglial regulation in an age-dependent manner. Furthermore, our data suggest that CB_2_R deletion contributes to microglial activity and might be crucial for microglial activation itself.

## 4. Materials and Methods

### 4.1. Animals

The generation of CB_2_R^−/−^ mice has been previously described [62]. C57BL/6J were originally obtained from a commercial breeder (Charles River) and bred in house. CB_2_R^−/−^ mice were bred homozygous and backcrossed to the C57BL/6J line every six generations to minimize the risk of genetic drift.

All animals were housed in specific-pathogen-free conditions in the main animal facility of the University of Bonn. After weaning, mice were housed grouped in standard laboratory cages, with an automatic ventilation system, and *ad libitum* water and food access, under 12 h light-dark cycle (lights on at 09:00 a.m.). Cages were monitored daily and bedding, water, and food were changed weekly. Experiments were carried out with male mice at the age of around 3, 12, and 18 months.

Care of the animals and conduct of the experiments followed the guidelines of the European Communities Directive 86/609/EEC and the German Animal Protection Law regulating animal research and were approved by the Landesamt für Natur-, Umwelt-, und Verbraucherschutz (LANUV NRW), Germany (AZ 84-02.04.2017.A231).

### 4.2. Behavioral Testing

A week before the first behavioral test, mice were single-housed and transferred to a room with a reversed light-dark cycle (lights off at 9:00 a.m.). Tests were interspersed with 7-day intervals. Groups with 3, 12, and 18 month-old mice were tested independently and analyzed using Ethiovision XT 8.5 and 13 (Noldus, RRID:SCR_000441). The experimenter was blind to the genotype.

### 4.3. Partner Recognition

Partner recognition (PR) paradigm was used to assess social memory. The test was performed in an open-field box (44 cm × 44 cm) containing a thin layer (about 1 cm) of sawdust. For three consecutive days, mice were allowed to explore the arena freely for 10 min, and habituate to the environment. On the test day, mice underwent two trials. In trial 1, mice were given 9 min to freely explore the arena containing an object (metal can) and a grid cage (diameter about 10 cm, height about 12 cm) with an unfamiliar C57BL6/J male partner mouse. The can and the cage were in opposite corners, each placed about 6–7 cm from the wall. Partner mice were approximately 10 weeks old. Interaction was noted when the mouse nose point was within 2 cm of the cage/object. The time spent on top of any of the objects was deducted from the interaction time. After trial 1, mice were returned to their home cages for 30 min (Figure 1) or 1 h (Appendix A). Sociability in trial 1 was calculated as follows: sociability (%) = Tp/(Tp + Tc) * 100, where Tp is the time of interaction with a partner mouse, and Tc is the time of interaction with the object.

The mean sociability value was tested with a one-sample *t*-test against the chance level. Values above 50% indicated that the mouse spent more time interacting with a partner than with an object. In trial 2 the metal can was replaced by a grid cage with a novel mouse and the test mouse was given 3 min to freely explore and interact with both caged mice. Preference for the novel mouse was calculated as: preference (%) = Tn/(Tf + Tn) * 100, where Tf is the time spent with the familiar mouse and Tn is the time spent with a novel mouse. A preference for the new partner was interpreted as evidence for social memory. To detect learning in each group, we analyzed if the preference for the novel partner deviated statistically from the chance level with a one-sample *t*-test against a hypothetical mean (50%). Mice with sociability ≤55% were excluded from the analysis. If partner mice showed any signs of aggression, they were excluded from the analysis.

### 4.4. 0-Maze (Elevated Zero Maze)

The zero maze consisted of a circular runway with a diameter of 47 cm and width of 5.6 cm, elevated 30 cm above the ground. It was divided into four equally sized compartments, two of which were enclosed by 24 cm high walls. Mice were allowed to explore the maze for 5 min. Light intensity was around 200 lx. Head-dipping in the open compartment and stretched-attend postures were counted manually as described previously [59].

### 4.5. Morris Water Maze

The Morris water maze (MWM) was used (Morris 1981) to assess spatial learning and memory. In this paradigm, mice learn to locate a submerged and invisible platform in a round basin filled with turbid water, based on spatial cues. The experiment consisted of three phases: acquisition (days 1–6), probe trial (day 7), and reversal phase (days 8–10). During the acquisition phase the hidden platform remained in a fixed location and animals swam four times per day from different entry points. Inter-trial interval time was 1 h. The cut-off time for each swim was 90 s. If the mouse located the platform within the time limit, then it remained on it for an additional 5 s before being taken out of the maze. Otherwise, after the time limit passed, the mouse was guided to swim to the platform and remained there for an additional 20 s. The decrease of the time required to find the hidden platform indicated spatial learning. In the probe trial, the platform was removed and the time spent in the platform-associated quadrant was measured for 90 s. For the reversal phase, the platform was placed into the opposing quadrant, thus necessitating a re-learning of the position. Similar to the acquisition phase, mice swam four times daily and the latency to the platform was recorded. One mouse that stayed close to the wall at all times was excluded from the analysis.

### 4.6. Organ Extraction

Mice were anesthetized and perfused transcardially with PBS. Brains were hemisected and one hemisphere was post-fixed in 4% *w/v* formaldehyde for 3.5–4 h on ice. Afterwards, left hemispheres were incubated overnight in 10% sucrose, followed by an overnight incubation in 30% sucrose. The hemispheres were then frozen in dry ice-cooled isopentane and stored at −80 °C. The hippocampus was dissected from the right hemisphere and snap-frozen in liquid nitrogen.

### 4.7. RNA Isolation and DNase I Digestion

Total RNA was isolated from PBS-perfused, right hemispheric hippocampi (n = 6 hemispheric hippocampi per genotype per age group) using the TRIzol^®^ protocol. Briefly, frozen tissue was homogenized in 1 mL or 800 μL TRIzol (Invitrogen, Camarillo, CA, USA). Tissue homogenates were centrifuged and mixed with 160 μL chloroform. RNA was precipitated with 400 μL ice-cold isopropanol, washed twice with 75% ethanol, and the resulting pellet was dried. Subsequently, RNA was incubated with 2 μL DNase buffer, 10U DNase I and RNase free water in a total volume of 20 μL for 30 min at 30 °C, followed by a DNase inactivation at 75 °C for 5 min. RNA samples were stored at −80 °C.

### 4.8. cDNA Synthesis

For cDNA synthesis, 1080 ng RNA was incubated for 5 min at 65 °C and then reverse transcribed at 42 °C for 50 min. A total volume of 20 μL included 4 μL first-strand buffer (Invitrogen), 2 μL 0.1 mol/l DTT, 1 μL 10 mmol/l dNTPs, 0,5 μL oligo(dT) 20 primer (Invitrogen), and 200 U Super-Script II reverse transcriptase (Invitrogen).

### 4.9. Quantitative Real Time PCR (qPCR)

Analysis by qPCR of cDNA samples was performed using a BioRad CXF384 Cycler and ThermoFisher TaqMan^®^ Gene Expression system. A 30 ng quantity of cDNA was used per reaction. A standard program was applied as follows: step 1 (1×) 95 °C, 10 min; step 2 (40×) 95 °C, 15 s and 60 °C, 1 min. TaqMan primer (all Applied Biosystems, Foster City, CA, USA): *hprt* (Mm03024075_m1), *tnfa* (Mm00443258 _m1), *il6* (Mm00446190 _m1), *ccl2* (Mm00441242_m1), *arg1* (Mm00475988_m1), *nos2* (Mm00440502_m1).

### 4.10. Immunohistochemistry and Imaging

Mouse brains were sectioned coronally at a thickness of 50 μm using a cryostat. Five dorsal hippocampal sections per mouse were stained as free-floating sections. Sections were post-fixed in 4% PFA for 2 h at room temperature (RT). After three washing steps with PBS, slides were blocked overnight at 4 °C in 10% *w/v* bovine serum albumin (BSA), 2% normal goat serum, and 0.5% Triton X-100 in PBS. Blocked sections were incubated with primary antibodies for 48 h at RT in the dark and, after several washing steps, incubated with secondary antibodies for 4 h at RT. Finally, slices were incubated with 0.1 μg/mL DAPI for 15 min and mounted on a slide using Fluoromount-G™ Mounting Medium. The following antibodies were used: Iba1 (AB_839504), CD68 (AB_322219), goat-anti-rabbit AF647 (AB_2535813), and goat-anti-rat AF488 (AB_2534074). High-resolution images were acquired with a confocal laser scanning microscope (Leica TCS SP8) using a 63x water-immersion objective lens (NA = 1.2). In each experiment, two z-stacks (about 30 µm; step size 0.5 µm; 0.18 µm/px; resolution 1024 × 1024 px) were acquired per mouse of the strata radiatum and pyramidale in the CA1 hippocampal region. Lipofuscin accumulation was measured as autofluorescence (576–640 nm).

### 4.11. Image Analysis

Quantitative cellular parameters were determined using ImageJ (FIJI ver. 2.0, and higher). Two z-stacks with at least 5 microglia/stack were analyzed for each animal. Stacks from 6 mice/genotype were analyzed per age group.

### 4.12. CD68 Area Fraction

The CD68 content was determined within each microglial soma. Briefly, maximum intensity projections of the CD68 channel z-stacks were generated using the ‘z project’ command. Images were binarized with the ‘threshold’ command. Mean grey value threshold was kept constant among all groups. The threshold used was determined as an average individual threshold of all WT images. The area fraction of CD68 signal was measured within each microglial soma using the ‘measure’ command.

### 4.13. Microglial 3D Reconstruction and Analysis of Microglial Branching

Microglial morphology was quantified using a custom-written ImageJ toolbox designed to reconstruct and analyze microglial cells, similar to previous studies from Plescher et al. (2018) and Schmöle et al. (2018) [41,60]. The toolbox consists of three ImageJ plugins for single-cell image generation, image segmentation, and cell analysis. Per group in each genotype, at least 50 microglial cells were selected in all z-slices of confocal z-stacks using the single-cell selection plugin by an investigator who was blind to the experimental conditions. The resulting single-cell images were segmented using the image segmentation plugin. An intensity threshold (algorithm: “Huang”) was calculated in an 8-bit converted, 0.5-fold scaled, and maximum-intensity projected copy of the original image. The threshold was applied to the unmodified original image. Segmented images were analyzed using the cell analysis plugin after applying a particle-filter (Length calibration = 0.3608 μm/pixel, Voxel Depth = 0.5 μm/voxel, minimum particle volume = 10,000 voxels). The microglial mean Iba1 intensity was determined as the mean intensity of all voxels in the original image that were positive in the particle-filtered, segmented image. The 3D microglial ramification index was defined as: cell surface area/(4π∙[((3∙cell volume)/(4π))]^(2/3)), which describes the ratio of cell surface to cell volume and serves as a sensitive measure for cell shape complexity. To determine the “Branch number” and “Tree length”, the segmented images, after particle filtering, were Gauss-filtered (Sigma XY = 1.0 and Sigma Z = 0.0), skeletonized using the Fiji plugin “Skeletonize3D” [63], and analyzed using the Fiji plugin “Analyze Skeleton” [63]. The polarity index indicates how equally the process tree is distributed around the cell soma. It was defined as the length of the vector from the center of mass of the microglial cell to the center of the convex hull around the microglial cell, normalized to the size of the convex hull: polarity index = vector length/($2\cdot \sqrt[{3}]{3\cdot spanned\;volume/\left(4\pi \right)}$).

### 4.14. Lipofuscin Analysis

Neuronal accumulation of lipofuscin was measured in the stratum pyramidale of the hippocampal CA1 region from a binarized max z-projection (7 image planes) with a defined start. Number of lipofuscin particles of a size >0.5 µm was counted using the ‘particle analyzer’ plugin in ImageJ. Density was calculated as the number of lipofuscin particles divided by area of selection. Lipofuscin levels were measured in the soma of single stratum radiale microglia using binarized maximum z-stack projections as described above for the CD68 area fraction with at least 12 cells per mouse.

### 4.15. Soma Size and Iba1 Intensity

The somas of microglia were manually delineated using the ‘polygon selection’ tool and saved as ‘regions of interest’ (ROI). Iba1 intensity and soma size were measured within each ROI with the ‘measure’ command. Soma size was measured in both Lipofuscin and CD68 area fraction experiments and both datasets were pooled together.

### 4.16. Statistical Analysis and Data Presentation

Microsoft Excel (v 16.43) was used for data analysis followed by statistical analysis and data visualization in GraphPad Prism version 7.0.0 and 9.1.2 for Mac, GraphPad Software, San Diego, CA, USA, www.graphpad.com. Figures were created in Adobe Illustrator (v 24.0.2). For presentation, representative images were post-processed in ImageJ (Fiji) to adjust brightness and contrast. All images within one experiment were adjusted the same way. Behavioral data were analyzed using Ethiovision XT 8.5 and 13 (Noldus, RRID:SCR_000441).

For datasets consisting of more than two groups with two independent variables (e.g., genotype and gender), two-way analysis of variance (ANOVA) was used followed by Sidak’s multiple comparison test. For MWM repeated measurement (RM) ANOVA was used. For PR, the mean of the group was tested against a theoretical mean (50) with one-sample *t*-test. For single microglia analysis, an outlier test was performed with ROUT = 5% prior to the analysis, with the detected outliers excluded. For expression analysis, an outlier test was performed with ROUT = 10% prior to the analysis, with the detected outliers excluded. Datasets with more than 20 points were depicted using a violin plot to precisely visualize the distribution of the data, whereas datasets with fewer than 20 points were depicted as scatter plots. Statistical significance was stated when p-value < 0.05 at a 95% confidence interval. Detailed results of statistical analysis are presented in Appendix A.

## Figures and Tables

**Figure 1 molecules-26-05984-f001:**
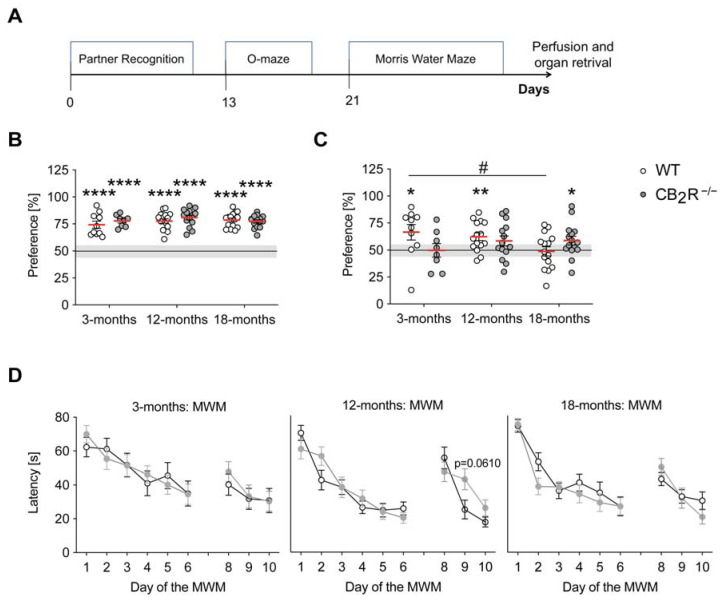
CB_2_R deletion has a moderate age-dependent effect on cognition. (**A**) Experimental timeline: Partner Recognition (PR), O-maze, and Morris Water Maze (MWM). (**B**) Sociability in the PR test was calculated as interaction time with a partner mouse over total interaction time. All groups were social as indicated by mean sociability >50%. (**C**) Preference for the novel partner (after 30 min separation) was calculated as time with the novel partner mouse over total interaction time. Each group was analyzed individually by one-sample *t*-test (hypothetical mean = 50). Significant difference from the 50% chance level indicated learning (one sample *t*-test against a hypothetical mean (50%): 3-months WT mice *p* = 0.0021). * *p* < 0.05; ** *p* < 0.01; **** *p* < 0.0001. Each point represents a single mouse. Red line indicates the mean value ± SEM. Line indicates a 50% chance level. Grey box indicates 5% variance around the chance level. Two-way ANOVA followed by Sidak’s multiple comparison test with # *p* < 0.05 significance between age groups within the same genotype. (**D**) Acquisition and reversal phase of the MWM. Panels from left to right: 3-, 12- and 18-months old mice. Decrease in the average latency was detected in all groups. RM ANOVA followed by Sidak’s multiple comparison test with exact *p*-value reported between genotypes within the same age group. WT mice—white circle; CB_2_R^−/−^ mice—grey circles. N = 14–15 mice/genotype/age group.

**Figure 2 molecules-26-05984-f002:**
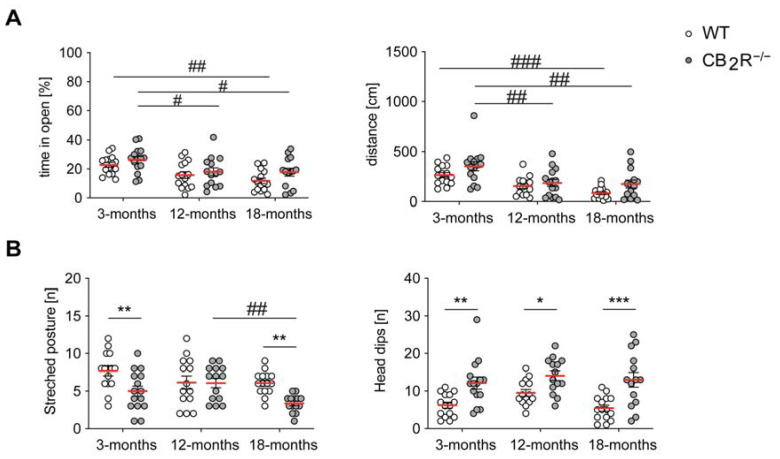
CB_2_R deletion results in a decreased anxiety phenotype in O-maze. (**A**) Left panel: % of time spent in the open compartment depended on the age and genotype of the mice. Decreased % of time indicates higher anxiety. Right panel: distance travelled in the open compartment depended on the age and genotype of the mice. Decreased distance indicates higher anxiety. (**B**) Left panel: number of stretched posture behaviors was dependent on genotype and age. Increased number of stretched postures indicates higher anxiety. Right panel: number of head dipping behaviors in the open compartment was increased in CB_2_R^−/−^ mice independent of age. Decreased number of head dips indicates higher anxiety. WT mice—white circle; CB_2_R^−/−^ mice—grey circles. N = 14–15 mice/genotype/age group. Each point represents a single mouse. Red line indicates the mean value ± SEM. Two-way ANOVA followed by Sidak’s multiple comparison test with * *p* < 0.05, ** *p* < 0.01, *** *p* < 0.001 significance between genotypes within the same age group; # *p* < 0.05, ## *p* < 0.01, ### *p* < 0.001 significance between age groups within the same genotype.

**Figure 3 molecules-26-05984-f003:**
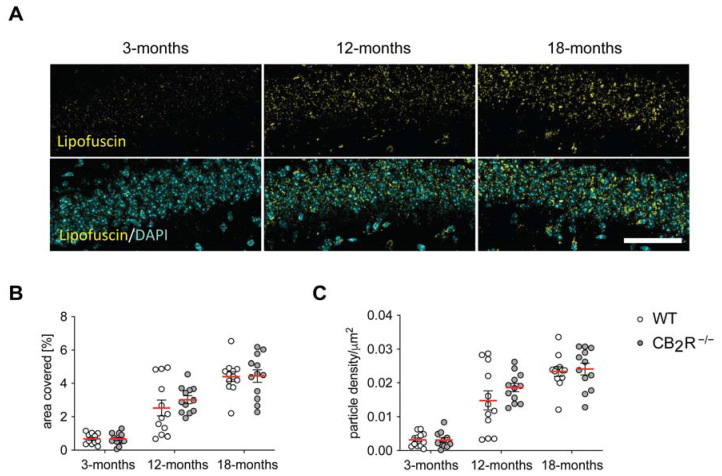
Accumulation of Lipofuscin in hippocampal pyramidal neurons during aging is not altered by CB2 deletion. Representative microscopy images of Lipofuscin accumulation in hippocampal pyramidal neurons with a scale bar of 50 μm (**A**). The area covered with lipofuscin (**B**) and the particle density (**C**) show a significant increase but is not different between WT and CB_2_R^−/−^ mice. WT mice—white circle; CB_2_R^−/−^ mice—grey circles. N = 6 mice/genotype/age group, two substacks per animal. Each point represents a single substack. Red line indicates the mean value ± SEM. Two-way ANOVA.

**Figure 4 molecules-26-05984-f004:**
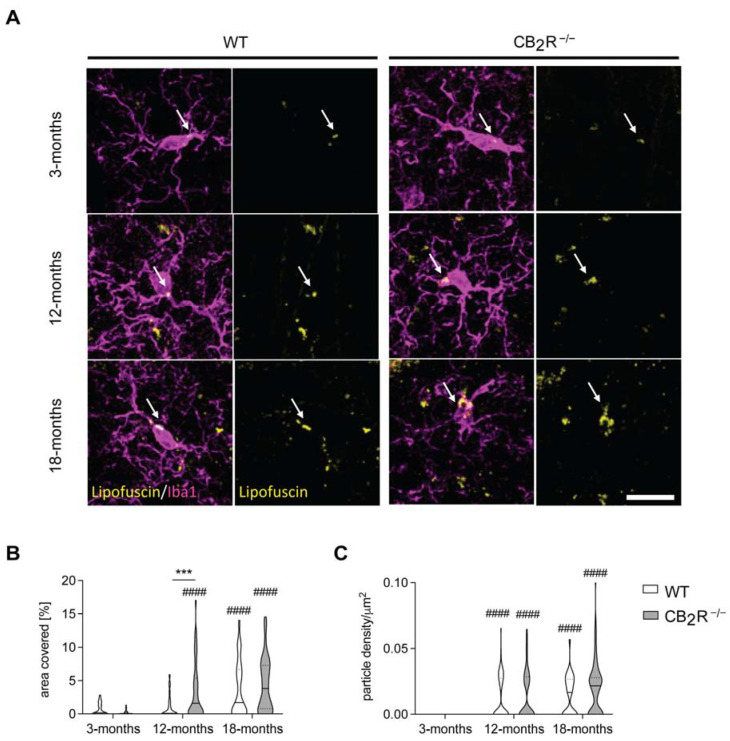
Accumulation of Lipofuscin in microglia is increased after CB2 deletion. Representative microscopy images of Lipofuscin accumulation in microglia in the stratum radiatum of the hippocampus. Scale bar represents 50 μm (**A**). The microglial somatic area covered with lipofuscin (**B**) and the particle density (**C**) show a significant increase with age in both genotypes. Microglia from CB_2_R^−/−^ mice show enhanced lipofuscin accumulation in comparison to WT mice, which resulted in an interaction effect. WT mice—white; CB_2_R^−/−^ mice—grey. N = 6 mice/genotype/age group. Data displayed as median (full line) with 25 and 75 percentiles (dotted lines). Two-way ANOVA followed by Sidak’s multiple comparisons with *** *p* < 0.001 significance between genotypes within the same age group; #### *p* < 0.0001 significance in relation to 3-months old group within the same genotype.

**Figure 5 molecules-26-05984-f005:**
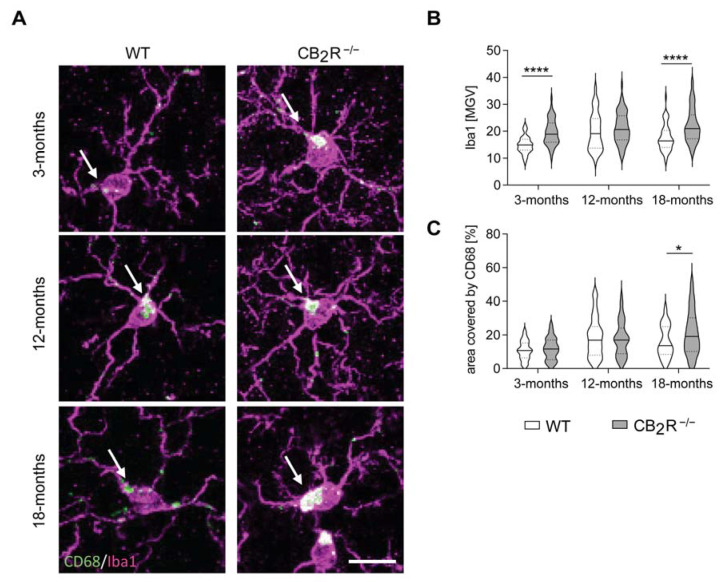
Iba1 and CD68 intensity is enhanced in CB_2_R^−/−^ microglia. Representative microscopy images from pyramidal microglia of 3-, 12- and 18- month old WT and CB_2_R^−/−^ mice with a scale bar of 10 μm (**A**). Iba1 intensity increased with age in both genotypes and was also enhanced in CB_2_R^−/−^ microglia when compared to WT microglia (**B**). CD68 expression was measured by area covered and increased significantly with age (**C**). Data displayed as median (full line) with 25 and 75 percentiles (dotted lines) (**B**,**C**). WT mice—white; CB_2_R^−/−^ mice—grey. N = 6 mice/genotype/age group. Two-way ANOVA followed by Sidak’s multiple comparisons with * *p* < 0.05, **** *p* < 0.0001 significance between genotypes within the same age group.

**Figure 6 molecules-26-05984-f006:**
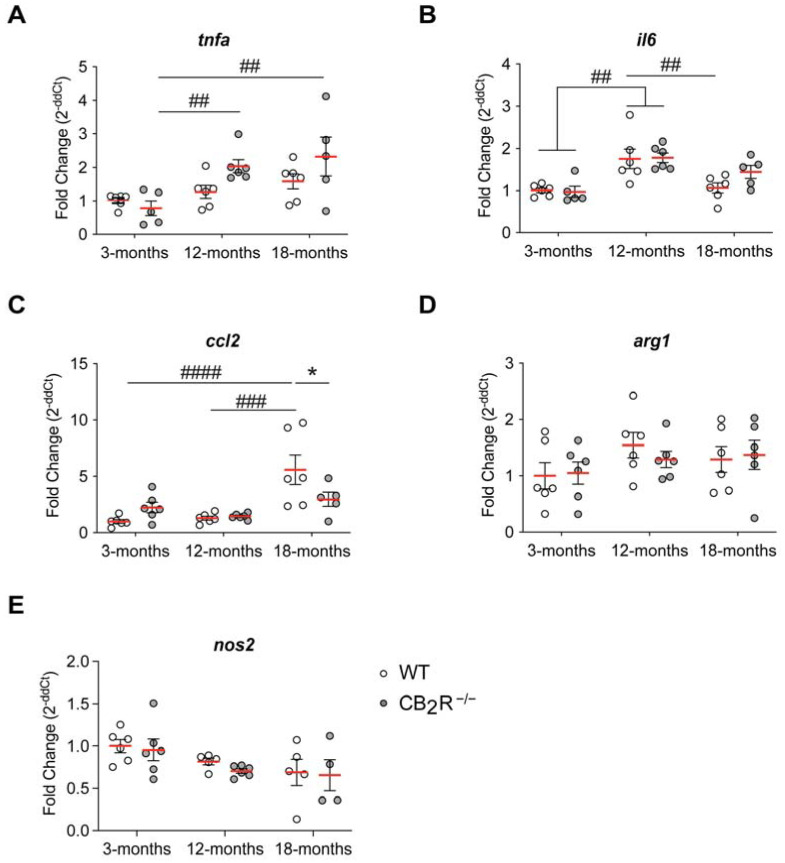
Age-dependent alteration in expression of *inflammatory mediators.* Expression of *tnfa* (**A**) and *il6* (**B**) in hippocampal tissue increases with age but does not differ between WT and CB_2_R^−/−^. Expression of *ccl2* (**C**) increases with age in WT, but not in CB_2_R^−/−^. Expression of *arg1* (**D**) did not differ between age groups and genotypes, whereas *nos2* expression (**E**) decreased with age. WT mice—white circle; CB_2_R^−/−^ mice—grey circles. N = 4–6 mice/genotype/age group. Each point represents a single mouse. Red line indicates the mean value ± SEM. Data were analyzed with two-way ANOVA followed by Sidak’s multiple comparison test, with * *p* < 0.05, significance between genotypes within the same age group; ## *p* < 0.01, ### *p* < 0.001, #### *p* < 0.0001 significance between age groups within the same genotype.

**Figure 7 molecules-26-05984-f007:**
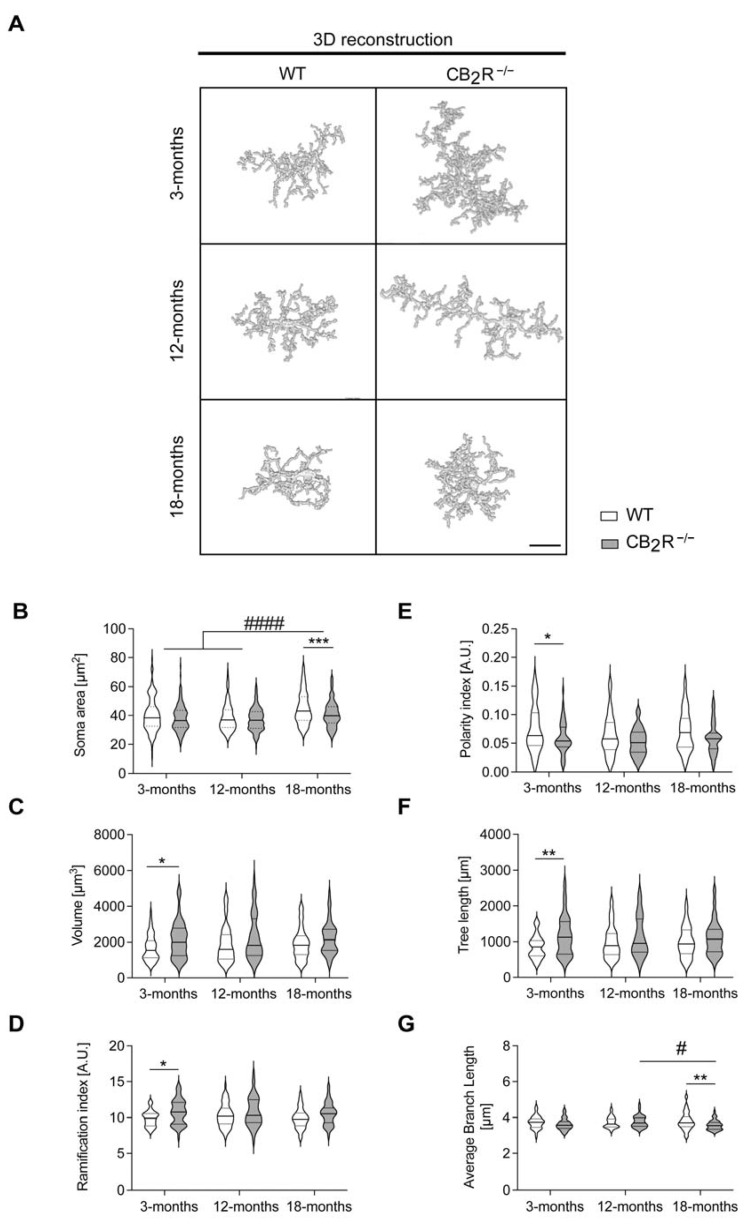
CB_2_R deletion changes hippocampal microglial morphology. Representative reconstruction images with a scale bar of 20 μm (**A**). Microglia morphology was analyzed by measuring the soma size (**B**), volume (**C**), ramification index (**D**), polarity index (**E**), tree length (**F**), and average branch length (**G**). Microglia morphology differs between CB_2_R^−/−^ mice and WT mice. Soma size increased with age in both WT and CB_2_R^−/−^ microglia. Data displayed as median (full line) with 25 and 75 percentiles (dotted lines). WT mice—white circle; CB_2_R^−/−^ mice—grey circles. N = 6 mice/genotype/age group. Two-way ANOVA followed by Sidak’s multiple comparisons with * *p* < 0.05, ** *p* < 0.01, *** *p* < 0.001, significance between genotypes within the same age group, # *p* < 0.05, #### *p* < 0.0001 significance between age groups within the same genotype.

## Data Availability

Datasets are available on request. The raw data supporting the conclusions of this article will be made available by the authors, without undue reservation, to any qualified researcher.

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
