# Peer review of "Cannabinoid Receptor 2 Alters Social Memory and Microglial Activity in an Age-Dependent Manner"

_molecules, 2021, doi:10.3390/molecules26195984_

Round 1

Reviewer 1 Report

Manuscript #molecules-1346279 describes an interesting study exploring the involvement of cannabinoid receptors type 2 in cognition and microglial activation upon aging. This study employed wild-type and CB2R-null mice. It concluded that "CB2 plays a role in cognition and microglial regulation in an age-dependent manner". An important question that was not addressed is that whether administration of CB2R agonist/antagonist would be able to reverse/enhance these effects or not. I suggest authors to include a selective CB2R agonist and an antagonist in their study. In addition to addressing this question, it would also fulfil the journal requirement which is "Studies must include bioactive synthesized or isolated compounds or probes". Other than the above-mentioned major issue, few minor language mistakes or typos should be corrected.  

Reviewer 2 Report

      In this manuscript, the authors aimed to examine whether and how CB2R deletion affects microglia response and cognitive function in young and aged animals.  Despite mice with gene deletion of CB2R showed an increased accumulation of lipofuscin in microglia and the accompanied morphological changes, the age-dependent cognitive decline seems to be genotype-independent. This is an interesting study to elucidate the role of CB2R in age-dependent inflammation and aging process. A number of questions need to be addressed:

  1. The increased inflammatory gene expression in aged animals is not correlated with the changes of microglia morphology. The reduced microglia function in CB2R-/- young mice might contribute to the decreased social memory, what is the explanation for the increased social memory in the CB2R-/- old mice compared to the WT controls?
  2. What is known about the role of lipofuscin accumulation and phagocytic function in microglia?
  3. Given that CB2R deletion can affect social memory and anxiety, it is possible that the CB2R deletion may alter the inflammatory gene expression in PFC and amygdala, in addition to hippocampus. What are the changes of the microglial morphology and function in these regions?
  4. Does CB2R deletion affect the expression of astrocytes?
  5. To better elucidate the microglia function, more anti- and pro-inflammatory genes should be examined.
  6. In Figure 1B, it is unclear what does the symbol represent, please indicate in the figure legend.
  7. In the legend of Figure 2A, it said the “increased % of time indicates higher anxiety”. The opposite should be true.
  8. The symbols representing the statistical differences in the figure legends are not correctly matched in many figures (Figs. 2, 4-7). Please make corrections.
  9. The sex of animals were not mentioned and it is unclear whether there are sex differences in the parameters examined. If both sexes are used, the number of male and female animals in each experiment should be stated.
  10. The differences between the constitutive CB2R knockout and the CB2R mediated signaling in microglia response during the aging process should be addressed in the Discussion. 

Reviewer 3 Report

This study investigated the relation of the CB2 receptor with the physiological aging of the brain, paying emphasis on age-induced changes in certain cognitive processes and on the role played by microglial cells in these changes. This is a follow-up study conducted by the same German group that has investigated before other elements of the endocannabinoid system in relation with the physiological aging of the brain. In this new study, authors used several behavioral tests conducted in mice, normal or deficient in the CB2 receptor, and analysis of morphological and biochemical markers of microglial cells in the hippocampus of animals at 3-, 12- and 18-months of age. The idea is interesting, the study is well-designed and executed, and the results add new pieces of information that will help to elucidate the contribution of the endocannabinoid system in the physiological aging of the brain. I have only a few minor comments that I would recommend to be considered in a revised version. They are listed below:

  1. Results presented in the Figure 1: the data presented in panels B and C should be assessed by a 2-way (genotype x age) ANOVA (with repeated measures if the animals at each of the different ages are always the same) followed by an appropriate posthoc test.

  1. Results presented in the Figure 2A: Perhaps an additional analysis of these mice in one specific motor test (e.g. rotarod, open-field, actimeter) would be interesting.

  1. Although the abstract clearly indicated that the study of microglial cells was carried out in the hippocampus, this is not so clear in the text and in some of the figures showing this information. The mention is simply to pyramidal cells, but these are also present in cortical structures.

  1. The format of the references is not homogeneous and requires correction.

  1. THC should be always written as Δ9-THC to differentiate this phytocannabinoid from its isomer Δ8-THC.

Round 2

Reviewer 1 Report

In my humble opinion, revised manuscript might be accepted for publication. 

Reviewer 2 Report

The authors have addressed all my concerns and no further questions raised.